# Gene Therapy for Regenerative Medicine

**DOI:** 10.3390/pharmaceutics15030856

**Published:** 2023-03-06

**Authors:** Hossein Hosseinkhani, Abraham J. Domb, Ghorbanali Sharifzadeh, Victoria Nahum

**Affiliations:** 1Innovation Center for Advanced Technology, Matrix, Inc., New York, NY 10019, USA; 2The Center for Nanoscience and Nanotechnology, Alex Grass Center for Drug Design and Synthesis and Cannabinoids Research, School of Pharmacy, Faculty of Medicine, Institute of Drug Research, The Hebrew University of Jerusalem, Jerusalem 91120, Israel; 3Department of Polymer Engineering, School of Chemical Engineering, Universiti Teknologi Malaysia, Skudai 81310, Johor, Malaysia

**Keywords:** tissue engineering, gene therapy, viral vectors, non-viral vectors, regenerative medicine, biodegradable polymers, nanoparticles

## Abstract

The development of biological methods over the past decade has stimulated great interest in the possibility to regenerate human tissues. Advances in stem cell research, gene therapy, and tissue engineering have accelerated the technology in tissue and organ regeneration. However, despite significant progress in this area, there are still several technical issues that must be addressed, especially in the clinical use of gene therapy. The aims of gene therapy include utilising cells to produce a suitable protein, silencing over-producing proteins, and genetically modifying and repairing cell functions that may affect disease conditions. While most current gene therapy clinical trials are based on cell- and viral-mediated approaches, non-viral gene transfection agents are emerging as potentially safe and effective in the treatment of a wide variety of genetic and acquired diseases. Gene therapy based on viral vectors may induce pathogenicity and immunogenicity. Therefore, significant efforts are being invested in non-viral vectors to enhance their efficiency to a level comparable to the viral vector. Non-viral technologies consist of plasmid-based expression systems containing a gene encoding, a therapeutic protein, and synthetic gene delivery systems. One possible approach to enhance non-viral vector ability or to be an alternative to viral vectors would be to use tissue engineering technology for regenerative medicine therapy. This review provides a critical view of gene therapy with a major focus on the development of regenerative medicine technologies to control the in vivo location and function of administered genes.

## 1. Introduction

Gene therapy is a very promising and attractive technology that involves the in vitro or in vivo introduction of exogenous genes into cells for biological and therapeutic purposes. Regardless of the final goal for experimental biology and gene therapy, the first key task is to enable the gene to internalise into the cell as efficiently as possible and to facilitate the expression for a long or short time period. Gene therapy technologies have been used for different methods of viral vectors, non-viral vectors, and physical technologies to deliver genetic materials to cells and tissues. Regenerative medicine is an interdisciplinary field that combines engineering and live sciences to develop techniques that enable the restoration, maintenance, or enhancement of living tissues and organs on biomaterials. Its fundamental aim is the creation of natural tissue with the ability to restore missing organs or tissue functions that the organism has not been able to regenerate in physiological conditions. In planning gene therapy strategies for regenerative medicine therapy, one must consider two major avenues: direct gene delivery in vivo using viral or non-viral vectors or in vitro cell-mediated gene therapy. In both cases, the aim is to deliver a therapeutic gene of a growth factor or cytokine into the target tissue. Gene therapy is a preferred technology over the addition of a growth factor to the cell as, typically: (1) the half-life of the selected growth factor is short; (2) a single administration is usually not sufficient to elicit a biological effect; (3) the quantities required are prohibitively expensive; and, (4) continuous protein production increases the likelihood that a desired outcome will be achieved. This review summarises regenerative medicine therapy based on gene therapy and suggests new areas of investigation that may help to resolve problems. There is a remarkable overlap in how scientists and researchers employ the terms “tissue engineering” and “regenerative medicine,” and thus, we use these terms together in this review as regenerative medicine.

## 2. Gene Therapy Systems

### 2.1. Viral Gene Therapy-Based Systems

Viral vectors are very promising in gene therapy technology because they have very high efficiency in gene transfection. They can also be delivered both locally and/or systematically. However, several deficiencies must be remedied before they are employed in clinical applications. One of them is an inherent nature to induce immune and/or toxic reactions. Currently, some replication-deficient viral constructs are being used for most gene therapies, but it is difficult to completely rule out the possibility that recombination of the viral gene will generate a pathogenic, replication-competent virus de novo. There is also a limitation in the molecular size of a gene that can be introduced into cells by the viral vector. Difficulty of immunogenicity [1], vector generation [2], carcinogenesis [3], restricted DNA packing capacity [4], and broad tropism [5] can severely eliminate their applications. In contrast to the above-mentioned issues, viral gene therapy has had clinical trials with recent approvals for treatment of head and neck cancer, melanoma, and lipoprotein lipase deficiency [6,7,8,9]. Various viral gene therapy formulations continue to be developed clinically, including adeno-associated virus (AAV), lentivirus, and retrovirus. Although viruses are the most commonly researched vector, because of continuing safety concerns, research has broadened for developing non-viral alternatives.

#### 2.1.1. Adenoviral Vectors

Adenovirus (Ad) was the first DNA virus that has been therapeutically developed in gene therapy, mainly due to its high gene transduction efficiency, well-defined biology, genetic stability, and ease of fabrication [10,11,12]. It is a class of a non-enveloped, double-stranded DNA virus containing 57 serotypes of human Ads, which forms seven species, A–G [10]. A serotype is determined by the capability of infection in cell culture to be neutralised through distinct antisera. Among various serotypes of adenovirus, Ad5 is regarded as the most frequently used. The adenovirus genome is flanked on both ends by inverted terminal repeats (ITRs), and it contains early transcriptional units (E1, E2, E3, and E4), as well as a late unit [13]. The early units are accountable for expressing non-structural proteins, while the late unit encodes for structural parts of the Ad virion. A fundamental challenge of adenovirus gene-based therapy is the high immunogenicity of an Adcapsid, which severely eliminates the multiple administrations of Ad in the same patient to stop the risk of anaphylactic shock or death [14,15,16,17,18,19]. This drawback is beneficial in vaccination and oncolysis applications [20,21,22]. Adenoviral vectors have not only been used against influenza and human immunodeficiency virus (HIV) [23,24], but also employed to trigger antitumoral immunity in the tumour microenvironment [21,25]. They have substantially advanced gene therapy as the most successful in vivo gene delivery, which has yielded high levels of the transgene expression [10]. Their safety issues have been recently addressed with the safest dosing and routes of administration [10]. In fact, more than 20% of all gene therapy trials consist of adenoviral vectors, making them the most frequently used viral vector-based gene therapy in clinical applications. Another excellent feature of adenoviral vectors is their ability to transduce quiescent and replicating cell populations as most cells in human tissues are both non-dividing and resistant to transduction [14]. Adenoviral vectors are also capable of carrying up to 8 kbp heterologous DNA in size, and resides episomally in the host nucleus, even though their DNA does not integrate into the host genome [26,27].

##### Barriers to Adenovirus-Mediated Gene Therapy

The biggest obstacles to the therapeutic use of adenoviral vectors in gene therapy are reactions with the immune system, vector packaging capacity, viral longevity, and contamination with a helper virus (HV). High-capacity adenoviral vectors (HC-AdVs) have the potential to address many of these limitations. HC-AdVs, also referred to as gutless AdVs or helper-dependent AdVs (HD-AdVs), are lesser immunogenic vectors stripped of all viral coding sequences, hence, causing a larger capacity for transgenic cloning sequences (36 kb) [28]. The immune reduction of HC-AdV not only prevents tissue damage and inflammation, but also increases the lifespan of the virus. One significant issue associated with the increased expression of HC-AdV is owing to previous immunisation against Ad. For this reason, optimisation of HC-AdV longevity via encoding novel tetracycline-dependent (TetOn)-regulatory elements is desirable [29]. For example, the regulation of HC-Ad-TetOn vectors with doxycycline indicates higher efficacy and regulation than constitutively active HC-AdV in the existence of an immune response [30]. The lack of viral genes can also lead to higher packing capacity. Nevertheless, even the presence of viral packaging machinery can have too little specific enzyme sites for transgene insertion. This drawback has been addressed through the development of specific HC-AdV possessing a transgene insertion site with the ability of maintaining tissue-specificity [31]. AdV-mediated gene delivery also suffers from contamination with a helper virus. This limitation can be overcome by HC-Ad (which is based on minimal HV contamination), Cre/loxP, and FLP/FRT systems. Although these approaches have been shown to be successful, the contamination levels of AdV for clinical studies can still be too high. An accepted strategy for reducing contamination in the production of HC-AdV involves self-inactivating HVs based on virally encoded recombinases. Thus, a unique HV (AdTetCre) in which its packaging signal is flanked by loxP sites could be excised through a chimeric MerCreMer recombinase encoded in the same viral genome [32]. The MerCreMer expression was modulated by a Tet-on inducible system, enabling the fabrication of titers with great numbers of infectious units (>10^10^) and very low HV contamination (<0.1%). Other issues related to the AdV-based gene therapy are viral transport and targeting, as well as cellular barriers. A recent study reveals that expression levels of AdV receptors do not correspond to their biological function or transduction efficiency [33]. An improved understanding of HC-AdV challenges may facilitate transduction yields and overall efficacy.

#### 2.1.2. Adeno-Associated Viral Vectors

Adeno-associated virus (AAV) is a small, non-enveloped, single-stranded DNA that belongs to the genus Dependovirus, the family Parvoviridae. This helper-dependent virus can transduce both non-dividing and dividing cells by delivering a primarily episomal transgene product [5,34,35]. It exists in many species, including human and non-human primates. To complete its replication cycle, AAV requires co-infection with a genotoxic agent or AdV or HSV. In a site-specific manner, AAV integrates into the q arm of human chromosome 19, where expression remains silent until the AAV lytic cycle is activated by a helper virus [36,37]. The AAV genome contains two genes, *rep* (replicase, required for viral genome replication) and *cap* (encoding the structural proteins), which are flanked by two short ITRs. During vector production, a gene of interest is incorporated between the ITRs in place of *rep* and *cap*, and the latter are provided in *trans* along with helper viral genes [38]. AAV has been considered to be a potential candidate in gene delivery because of its low immunogenicity, capability of infecting both dividing and quiescent cells in vitro and in vivo, and site-specific integration [36,37,39]. A recent research by Suh and colleagues has developed a protease-activated AAV2 vector (Provector) platform to decrease off-target viral delivery in metastatic epithelial ovarian cancer [40]. More than 99% of injected AAV2 has been cleared from the blood in 1 h, while only 10% of the injected Provector is still detected in the blood after 24 h. The Provector also indicates the effective off-targeting of healthy tissue, specifically the liver, where viral delivery is less than 1% of AAV2. 

##### Barriers to Adeno-Associated Virus

One of the major challenges for AAV-based gene therapy is large-scale production and cost. For instance, the price of related drugs for in vivo gene therapy, Glybera and Luxturna, is extremely high [41]. The small transgene capacity of AAVs that can only carry approximately 4.7 kb heterologous DNA drastically restricts their gene delivery applications [14,22,42]. Another obstacle might be the interaction of recombinant AAVs protein capsid, its DNA genome, and the protein product of the transgene with host immune systems at multiple layers [43,44]. Before gene expression begins, the single-stranded AAV DNA should be converted into double-stranded DNA, which is a time-consuming process for in vivo studies [45]. This drawback can be addressed through the modification of the AAV vector using HSV [46]. Despite these limitations, several types of AAV vectors derived from the AAV9 serotype have been shown to cross the blood-brain barrier and confirm the ability of transducing cells of the central nervous system [47,48,49]. AAV vectors have also been extensively investigated for diverse pre-clinical and clinical trials, including Parkinson disease, giant axonal neuropathy, achromatopsia, choroideraemia, Crigler–Najjar syndrome, Pompe disease, and HIV infections [41].

#### 2.1.3. Retroviral Vectors

Retroviruses are enveloped viruses with single-stranded positive-sense RNA genome of 7 to 12 kb [50]. To generate structural enzymes and proteins for viral metabolism, three large open reading frames (ORFs), *gag*, *pol*, and *env*, are required. The functions of these three genes are explained in lentiviral vectors. The retroviral genome has two long terminal repeats (LTRs) at 5′ and 3′ extremities that act as promoters and adjust the expression of *gag*, *pol*, and *env*. The RNA genome is converted into double-stranded DNA by reverse transcriptase when the capsid is inside the host cell. Finally, double-stranded DNA is delivered to the nucleus and integrated into the host cell genome. The 5′ LTR sequence contains a strong promoter region including a highly active initiator sequence and many *cis* elements for transcription-factor binding. On the other hand, the 3′ LTR functions as the polyadenylation and termination site for all viral ORFs [51]. For gene therapy applications, deletion of three ORFs (*gag*, *pol*, and *env*) and other non-related sequences allows retroviral vectors to carry up to 8 Kbp heterologous DNA. Most of the retrovirus vectors employed in gene therapy are derived from a murine Moloney leukemia virus (MLV). Nonetheless, MLV vectors cannot efficiently transduce, slowly dividing or non-dividing cells. The genes required for viral infection are supplied in *trans* and are expressed in several plasmids during retroviral generation. 

##### Barriers to Retroviral Vectors

Integration of the retroviral DNA genome (provirus) can possibly disrupt the gene structure, varying its function/transcription, eventually causing oncogenesis. The genomic DNA is completely accessible, as its integration can take place in replicating cells [52]. Retroviruses also suffer from the absence of or very low pre-existing immune population as well as low immunogenicity. They are not able to infect non-dividing cells [53]. This severely eliminates the application of retroviral vectors-based gene therapy, because most cells in the human body are dividing or quiescent [54]. Retroviruses are still regarded as the most suitable vectors for ex vivo gene therapy despite potential limitations related to inactivation of tumour suppressor functions or insertional activation of oncogenes.

### 2.2. Non-Viral Gene Therapy-Based Systems

Non-viral vectors are synthetic polymeric materials which have been widely used in gene therapy technology. A cationic liposome, Lipofectin^TM^, is commercially available for gene transfection. Calcium phosphate co-precipitation is another method for the formation of DNA nanoparticles. A co-precipitate of plasmid DNA with calcium phosphate is applied to cells for transfection [55,56,57,58,59,60]. However, the transfection efficiency is not high, and further material improvement is needed. Polycation technology based on complex-cell interaction is predicated on the simple and nonspecific electrostatic force, and more enhanced and cell-specific cell transfection are not always expected. The study of enhanced gene delivery based on new technologies of materials is underway [61,62,63,64,65,66,67]. One of the attractive technologies is based on cell receptor-mediated uptake [68]. There are several researches on the use of non-viral gene therapy-based systems in combinational technology of polymeric biomaterials and nanomaterials, which is just beginning to make an impact [69,70,71,72,73,74,75].

#### 2.2.1. Challenges of Non-Viral Gene Therapy 

The possible degradation of the therapeutic gene by endonucleases in the extracellular space and the physiological fluids is a major safety concern to non-viral DNA delivery. Encapsulation of the DNA in a nano-carrier can efficiently address this issue, since it prevents degradation by serum endonucleases and increases blood circulation time [76,77,78,79,80,81,82,83,84,85]. The example of nano-carriers are polyplexes (prepared through the condensation of negatively charged plasmid DNA by cationic polymers) [86,87,88,89], zwitterionic lipids [90], as well as a combination of cationic lipids, neutral lipids, and DNA [91]. In recent years and in close collaboration with the School of Pharmacy of Hebrew University of Jerusalem in Israel, we have developed a new class of biodegradable polycations capable of complexing and transfecting various genes to many cell lines in relatively high yields. Different polycations were prepared starting from various natural polysaccharides and oligoamines having two to four amino groups. These cationic polysaccharides were prepared by reductive amination of the oligoamine and periodate-oxidised polysaccharides. Although most of these cationic conjugates formed stable complexes with plasmid DNAs as determined by ethidium-bromide quenching assay [56], only the dextran-spermine-based polycations were found to be highly effective in transfecting cells in vitro as well as in vivo [55,56,57,59,60,61,62,63,64]. 

The second barrier to successful in vivo delivery of non-viral vectors is selective accumulation at the target tissue, cellular internalisation, and endosomal escape. The interactions of delivery systems with non-target cells can significantly affect delivery. For instance, several liposomal and small interfering RNA (siRNA) delivery systems interact with serum lipoproteins [92,93]. Conversely, undesirable aggregation can occur in particles with high positive surface charges [94,95,96,97,98,99,100,101,102,103,104,105,106]. A suitable method for decreasing non-specific interactions involves mostly the use of polyethylene glycol (PEG) that protects the surface of delivery carriers [107]. The need for in vivo delivery is a major challenge specifically in terms of siRNA. It is difficult for messenger RNA (mRNA) to diffuse across cell membranes due to its negatively charged, hydrophilic, and large size properties. Therefore, chemical modifications, using nano-carriers and direct injection, are assumed to be essential for therapeutic delivery of mRNA to the cytosol of target cells [76]. As with other nucleic acids such as DNA, mRNA, and short double-stranded RNA, Perfect mRNA delivery-based gene therapy mRNA is required to prevent degradation by extracellular and intracellular endo- and exo-ribonucleases [108,109], escape immune detection [110,111], endo/lysosomal escape [112,113], avoid nonspecific interactions with non-target cells or biomolecules, prevent renal clearance, permit extravasation to reach tissues of interest, and enhance cell entry [114]. Table 1 summarises the advantages and disadvantages of viral-mediated and non-viral-mediated gene therapy.

## 3. Regenerative Medicine Based on Gene Therapy

Regenerative medicine is recognised as an interdisciplinary field of research with the potential to repair or replace tissues, organs, or cells damaged by age, injuries, trauma, and disease. It encompasses numerous in vitro, ex vivo, and in vivo techniques that may require the transplantation of stem, differentiated, or engineered cells, alone or in combination with tissue engineering, delivery of therapeutic agents, and biomaterials scaffolds for enhancing the function of the host environment or reprogramming tissue and cell types [115,116,117,118,119,120,121,122,123,124,125]. The past two decades have witnessed diverse tissue engineering and regenerative medicine [126,127,128,129,130,131,132]. The U.S. Food and Drug Administration (FDA) has approved products, ranging from biologics (e.g., Apligraf, Carticel, Cord blood) and biopharmaceutics (e.g., Regranex, Osteogenic protein-1) to cell-based medical devices (e.g., Celution, Dermagraft) [116]. Biomaterials, stem cell, molecular imaging, and gene therapy approaches are substantially connected to regenerative medicine, since they aim to repair or replace damaged tissues [115,133,134,135,136,137,138].

### 3.1. Tissue Engineering Approach

Gene therapy in regenerative medicine can be achieved usually through two routes: (1) in vivo delivery (gene is directly introduced to site of interest) and (2) ex vivo delivery (gene is inserted into a cell that is subsequently transferred to the site of tissue damage in vivo). Despite low costs and being an easier procedure, targeting the specific tissue with direct control over their activities is difficult using in vivo gene therapy [139,140,141,142,143,144,145,146,147,148]. Otherwise, ex vivo gene delivery (also known as cell-mediated gene delivery) provides better control of bone regeneration [149]. Ex vivo gene delivery is properly matched with tissue engineering approaches, as it enables the removal, expansion, modification, and reinsertion of cells to site of injury [150]. However, this strategy is expensive, since autologous cells and genetically modified cells must be vastly tested prior to reinsertion [151]. Successful ex vivo delivery also requires the proper expansion and transduction of primary cell populations. For instance, adenovectors possess a rich level of transduction efficiency, whereas lentiviral vectors indicate lower level of transduction efficiency for murine cells as opposed to other mammalian cells [152]. Gene therapy can be classified into viral or non-viral gene delivery. Due to their high efficiency, viral vectors are considered as promising candidates in clinical trials [76]. However, safety concerns, difficulties in fabrication, as well as the risk of insertional mutagenesis that may cause leukaemia and oncogene transactivation are associated with their use [76,153]. Non-viral vectors ensure higher safety, lower immunogenicity, temporal transfection, and ease of fabrications in regenerative medicine areas compared to viral gene delivery [76]. In this section we extensively review the currently applied viral or non-viral gene therapy vectors using in vitro, ex vivo, and in vivo approaches.

#### 3.1.1. Angiogenesis

Angiogenesis is the formation of new blood vessels from the existing vasculature, which is organised by a vast number of angiogenic inhibitors such as caplostatin, vasostatin, alphastatin, and thalidomide, as well as angiogenic cytokines/factors, such as vascular endothelial growth factor (VEGF), basic-fibroblast growth factor (bFGF), and hepatocyte growth factor (HGF) [154]. This concept was first introduced by Folkman in 1971, who hypothesised the prerequisite of angiogenesis for tumour growth [155]. Angiogenesis is imperative in various physiological functions, such as placental growth, embryogenesis, and tissue regeneration, as well as pathological conditions such as cardiovascular/ischemic diseases, tumour growth, and rheumatoid arthritis [156]. Therefore, inhibition of angiogenesis (anti-angiogenesis) can play a vital role in the treatment of various tumour diseases. To target the underlying mechanism of angiogenesis, a wide variety of inhibitors have been developed, such as VEGF [157,158] and extracellular matrix (ECM) degradation [159,160]. This has resulted in the first-generation clinical trials of VEGF gene therapy in peripheral artery disease (PAD) and coronary heart disease (CHD) patients [161,162]. Despite pre-clinical and early clinical success, the obtained results have not been encouraging, which has further led to the development of bevacizumab and pegaptanib as promising anti-angiogenesis approved by the FDA [163,164]. A critical issue in brain tumour gene therapy is neurologic damage caused by traditional intracerebral injection and insufficient transgene expression as a result of the blood-brain barrier. Although non-viral gene therapy employing ultrasound-targeted microbubbles has been encouraging, no targeted gene expression and high-efficient approach has been developed. An angiogenesis-targeting and non-viral gene delivery method can arise for effective cancer therapy without causing any neurological damage via active targeting ability. To achieve this, Yeh and colleagues utilised the firefly luciferase gene (pLUC) and HSV type-1 thymidine kinase/ganciclovir (pHSV-TK/GCV) gene to fabricate a VEGF receptor 2 (VEGFR2)-targeted and cationic microbubbles (VCMBs) combined with transcranial focused ultrasound (FUS) for transient gene therapy in brain tumours [165]. This multifunctional system not only indicates great affinity for cancer cells and great DNA-loading efficiency, but it also improves gene delivery in the targeted tissues without affecting normal brain tissues. The delivery of both pLUC and pHSV-TK genes into the tumour cells has increased using this carrier. Compared with the direct injection (40.1 ± 4.3 mm^3^), tumour growth is efficiently inhibited in the pHSV-TK/GCV system using VCMBs in the presence of FUS (9.7 ± 5.2 mm^3^) (Figure 1). Anti-angiogenic therapy through the stable expression of exogenous wild-type p53 (wt-p53) gene can lead to tumour therapy by the direct interaction with VEGF promoter or indirect pathways [166]. Co-delivery of an anticancer drug and gene into the same cancer tissues can provide improved therapeutic treatments. For instance, the targeted co-delivery of wt-p53 and candesartan to angiotensin-overexpressed tumour cells demonstrates potent anti-tumour activity with high tumour-targeting capacity [167].

Ongoing angiogenic gene therapy trials reveal notable improvements in cardiovascular diseases, such as PAD and ischemic heart disease. Millions of lives have been affected by PAD worldwide, and existing treatments have not been completely effective. For example, a randomised controlled trial (RCT) has not shown a constant efficiency of the neovasculature in PAD patients, while it has indicated vascular growth effects of Ad-mediated VEGF-A and plasmid gene therapy [168,169,170]. Although a RCT analysis of cured patients has not exhibited a steady benefit of angiogenic gene therapy [171], Neovasculgen, as a plasmid encoding VEGF-A165, has been clinically approved in Russia for PAD therapy [172]. A combination of Angiopoetin (Ang)-1 and VEGF-A can remarkably enhance the clinical benefits and functionality of induced neovessels. Nonetheless, functional enhancements have not been observed in fibroblast growth factor-1 (FGF-1) plasmid in large RCTs TAMARIS [173] and TALISMAN [174], as well as adenoviral Hif-1a/VP16 [175]. Despite major unsuccessful clinical trials in PAD patients, several promising features such as peak walking time, improved tissue oxygenation, reduced amputation rates, and ulcer healing have been reported by employing HGF [176,177,178]. The current RCT analysis with approximately 200 patients aims to evaluate the influence of a NL003CLI-II-plasmid expressing two isoforms of HGF on ulcer healing. Currently, four types of angiogenic gene therapy are under clinical trials in CAD patients. Two of them use VEGFs for the treatment of myocardial ischemia, and the other two utilise intracoronary adenoviral FGF-4. KAT301 analysis evaluates VEGF-D in a small number of patients with no revascularisation opportunities and refractory angina, where increased doses of endocardial Ad-injections with electroanatomical targeting were applied. Three and twelve months after the therapy, O-radiowater-PET will be used to determine the absolute myocardial blood flow [179]. VEGF-D stimulates both lymphangiogenesis and angiogenesis, and it is considered as a potential vascular growth for refractory angina therapy, since it has not been previously investigated in humans [180]. Another angiogenic gene therapy is VEGF-A (VEGF-A116A) that uses all three main isoforms of humans [181].

#### 3.1.2. Bone Regeneration

Bone regeneration is a uniquely complex biological process that contains endochondral ossification and intramembranous ossification. The former process is responsible for long bone formation, while the latter plays a key role in the flat bones of the skull, clavicle, and mandible. Gene therapy strategies for bone repair and/or regeneration are of great importance in regenerative medicine, since they enable highly required enhancements in bone regeneration. They are based on the regeneration of cellular and biochemical composition of osseous tissue, as well as bone repair and development processes. Gene therapy for tissue regeneration has been employed for the delivery of angiogenic factors such as VEGF, bone morphogenetic proteins (BMPs), microRNAs, Wnt proteins, LIM-domain proteins (LMPs), and osteogenic transcription factors [150]. Selection of these factors is highly dependent on the choice between the endochondral ossification route and intramembranous ossification. The completed studies in this field are listed in Table 2.

##### Viral Gene Delivery

Primary studies of bone regenerations have primarily centred on recombinant human BMPs (rhBMPs). Recent evidence has demonstrated serious concerns in the use of rhBMPs, such as high cost, limited efficiency, and low stability. These issues can be addressed by the introduction of viral gene therapy and BMP. Liberman and co-workers employed adenoviral vectors of bone marrow-derived mesenchymal stem cells (MSCs) to evaluate the healing ability of rhBMP-2 and BMP-2 in a critical-size femur segmental defect. Compared with the protein alone, BMP-2 mediated MSCs (as gene delivery vehicles and therapeutic agents) show excellent bone formation in rat models [204]. The synergic effect of BMP combined with gene transfer for bone regeneration is also confirmed by other studies [205,206]. A wide range of gene therapy studies based on ex vivo methods employ MSC derived from bone marrow, fat, or muscle-derived stem cells in association with viral or non-viral vectors [207]. Promising results have been reported in advance of human clinical trials, using both small-animal models, such as rabbits [208], rats [182], and mice [183], as well as large-animal models, such as pigs [184], goats [209], and horses [210], in which bone-marrow derived MSCs or dermal fibroblasts are used as carriers for adenovirus delivery of *complementary DNA (cDNA*) of BMPs into cranial defects, long-bone defects, and sites of osteonecrosis of the hip. These results significantly highlight the efficacy of MSCs for ex vivo bone gene therapy areas. Bone regeneration-based ex vivo gene therapy can also be achieved through lentiviral and retroviral vectors. For example, the osterix gene transfer to mouse bone marrow-derived MSCs using retroviral vector leads to almost complete healing of calvarial bone defects in a mouse model [186]. In other research, BMP-4 delivery with lentiviral vector caused complete defect healing in a rat model [211]. Since lentiviral vectors are able to integrate into the host genome, they enable prolonged gene expression and highly efficient gene delivery. In an attractive method, a ‘same-day’ ex vivo gene therapy was employed to lentivirally transduce rat bone marrow cells intraoperatively [185]. This strategy revealed bone healing and higher bone volume in a rat femoral defect. The delivery of BMP-2 with lentiviral vectors generates a higher quality bone with a longer expression time in a rat segmental defect model compared with adenoviral gene delivery [212]. Prior to transduction with lentivirus, the bone marrow is fractionated to isolate the buffy-coat. In vivo gene therapy using adenoviruses delivering BMP-encoding cDNA [213] exhibits remarkable outcomes in bone regeneration of rats [214] and rabbits [215,216], but not in a sheep model [217]. Recombinant AAV-coated allografts are another innovative approach for in vivo bone regeneration that aims to utilise osteoclastogenic and angiogenic signals to cortical surfaces of allografts to show their revitalisation. However, the lack of vascularisation and remodelling ability of allograft raise concerns for clinical studies. To address this, Schwarz and colleagues employed transgenes encoding VEGF and a receptor activator of nuclear factor kappa B (NFκB) ligand in a mouse model that showed a formation of new bone collar around the graft [218]. The same group further reported the efficiency of this method in self-complementary rAAV-coated allografts exhibiting a 3-fold enhancement of graft bone volume than that of autografts [188]. Lentivirus and retrovirus hold potential bone regeneration applications, since they refrain the immune response evoked by adenoviruses and provide sustained expression of the osteogenic transgene [149]. For example, retrovirus carrying BMP-2/4 gene indicates a slow healing rate and an excessive generation of ectopic bone in a femoral fracture in a rat model [189]. The healing time and massive bone regeneration are further enhanced through the delivery of the cyclooxygenase-2 (COX-2) gene by retrovector [190]. Gene expression was specifically delivered to the defect site in both examples, showing the safety of retrovirus vectors.

##### Non-Viral Gene Delivery

Considerable interest has been placed on the use of non-viral bone regenerations, mainly because of their abilities to not only reduce the formation of undesirable bones, but also to decrease the amount of newly generated bones. They induce lower immune responses than viral vectors, which is advantageous in regenerative medicine or tissue engineering applications. Naked DNA mediated gene delivery is a desirable approach that enables ex vivo and in vivo delivery in regenerative medicine. In one of our pioneer research, MSC was seeded onto a gelatin-plasmid DNA of BMP-2 scaffold, and cultured by perfusion and static techniques in a rat model [191]. The poor transfection efficiency of naked DNA was highly improved by the cellular component enhancements when cultured by perfusion method. Although this method can heal bone defects in combination with MSc and BMP-2, concerns are still associated with their low efficiency and clinical use. Liposome delivery is an alternative method that provides higher efficiency of non-viral gene delivery. Non-viral BMP-2 gene delivery using liposome-mediated transfection leads to the favoured healing of critical-size bone defects ex vivo [203]. When implanted in a critical-size segmental bone defects, FGF-2 transfected MSCs show improved capillary regeneration and bone formation in a rat model [219]. A similar strategy was employed with rabbit MSCs transfected with plasmid DNA non-virally to deliver the osterix gene in a rabbit model. The results of this research indicate the effectiveness of osterix gene in bone regeneration [193]. In an interesting procedure, a BMP-6-encoding plasmid was nucleofected with porcine adipose tissue-derived MSCs [195]. This approach, that requires the electroporation (a high-voltage electrical current-mediated gene transfer) to non-virally deliver cells in vitro [220], successfully heals vertebral bone defects in a rat model. A method should demonstrate promising in vivo results to investigate the pre-clinical and clinical applicability of non-viral gene therapy for bone regeneration. An attractive method to enhance the structural integrity of new bone and increase the poor gene delivery is gene-activated matrix (GAM), which essentially contains a biodegradable scaffold including a plasmid DNA [149]. Proof of principle was first established in large segmental defects using a GAM-based plasmid DNA approach, in which an extensive amount of osteogenesis was successfully stimulated in an adult rat femur [197]. Effectiveness of in vivo gene delivery was subsequently reported with GAM technology using hydrogel as a substrate [221]. To achieve higher efficiency, cellular sonication (also known as sonoporation) has revealed the ability of transfecting cells with pDNA [201]. Ultimately, this technique involves increasing cell membrane permeability by an ultrasound. Using this enables the expression of the BMP-6 transgene and the transfection of 40% of the cells in pig cranial and tibial defect, resulting in complete fracture repair [200,201].

#### 3.1.3. Gene Therapy for Cartilage Regeneration

Gene therapy enables local expression of therapeutic gene products for articular cartilage therapy; however, the defects may result in osteoarthritis due to the limited self-healing capacity of the articular cartilage [222]. Compared with intra-articular recombinant growth factors, gene therapy might allow a sustained delivery of gene products for longer times [223]. Currently, used gene therapy procedures in cartilage regeneration are gene addition (augmentation), replacement (correction), silencing (alternation), and reprogramming (editing) through ex vivo or in vivo approaches that can lead to desirable outcomes [224,225,226]. A variety of viral and non-viral vehicles using ex vivo [227,228] and in vivo [187,229] procedures have been employed in cartilage repair via treating chondral (damage of articular cartilage positioned at the end of the bones) and osteochondral (damage of cartilage and neighbouring bone) defects. For example, suitable cartilage regeneration was achieved ex vivo by the administration of a bone marrow aspirate, including AdVs, delivering transforming growth factor B1 (TGFB1) to chondral defects in a sheep model [230]. Similarly, the use of a bone marrow aspirate includes AdVs and genetically modified chondrocytes delivering insulin-like growth factor I (IGF-I) [231], BMP2, or IHH [228]-enhanced cartilage repair in osteochondral defects in rabbits. In vivo gene transfers have also shown improved cartilage healing at 16 weeks for AdV carrying IL1RN and IGF1 in horses with chondral defects [229], and rAAV delivering SOX9 in rabbits with osteochondral defects [232]. Cucchiarini and Madry report that the rAAV-mediated IGF-I overexpression can stimulate the proliferation and matrix synthesis in vivo, causing osteochondral defects healing in the knee joints of rabbits [233]. IGF-I improved runt-related transcription factor-2 (RUNX2) expression levels, and the regeneration of the subchondral bone layer in the defects. Early attempts to treat rheumatoid arthritis in humans highlight the efficiency of gene therapy in regenerative medicine [234]. Ex vivo administration of a retroviral vector carrying IL1RN cDNA to human autologous synovial fibroblasts, followed by their intra-articular injections into the metacarpophalangeal joints, lead to a lower swelling and pain in patients with rheumatoid arthritis [234]. This strategy has since been adopted by various institutes/companies to deliver TGFB1 cDNA and rAAV-IL1RN to patients with rheumatoid arthritis [235,236].

From a clinical point of view, the size of the cartilage defect and its anatomic location tested in animal models should be similar to those of humans. A minimum of a one year study is required to evaluate cartilage healing in large animal models. This time duration ensures the durability of the therapeutic repair, as well as potential local and systemic toxicities associated with the viral and non-viral gene therapy techniques. The precise amount of the administrated dose that can directly alter the cartilage regeneration should also be carefully investigated [224]. After several years, the quality of cartilage regenerations in clinical studies in the case of both viral and non-viral gene therapy procedures should be evaluated [237,238,239]. To address these issues, a detailed knowledge of gene therapy approaches, tissues of interest, development of small vectors (with the suitable penetration ability into the cartilaginous area) [240], vector decoys [241], and engineered viral vectors [242,243,244] reveal remarkable outcomes.

#### 3.1.4. Cardiac Regeneration

Cardiovascular diseases are considered as a major cause of death. Damage to the heart results in the depletion of cardiomyocytes, which are further replaced by fibrotic scar tissue due to the limited regenerative capacity of cardiomyocytes in the human heart [245]. The loss of cardiomyocytes in the heart leads to reduced pump function and circulatory efficiency that ultimately cause fatal heart failure [246]. One therapeutic approach to repair the damaged heart and enhance heart function is gene therapy. Genes can be delivered to the heart through several strategies, in which myocardial injection, intracoronary perfusion, and atrial epicardial gene painting are the most extensively employed methods in large animal models. Intramyocardial cardiac gene delivery allows a high local concentration of the vector, because it contains the direct injection of the virus. The catheter-based origin of this approach provides minimally invasive surgery. However, transgene expression is restricted to the injection site and is also heterogeneous [247]. Intracoronary perfusion can be classified into anterograde intracoronary injection and retrograde venous injection. Anterograde intracoronary injection is an efficient and safe method successfully used in humans [248,249]. Nonetheless, the rapid transit of the vector through the coronary arteries is not desirable in cardiac applications. Retrograde venous injection has the advantage of hampering dilution of the gene construct in blood, while it may cause coronary occlusion [250]. Another delivery approach that can be exploited for targeted delivery to the atria is epicardial gene painting that involves brushing a solution, comprising a polymerisation mixture, the vector, and a diluted protease, onto the atrial epicardium. The polymerisation mixture provides higher vector–tissue contact time, and the diluted protease causes transmural penetration of the vehicle. However, the effectiveness of the injection is limited to the target area in which the solution spreads [251]. Gene therapy can be potentially adapted to restore dystrophin expression in Duchenne muscular dystrophy (DMD), an X-linked inherited disease caused by mutations in the gene coding for dystrophin that leads to a life-threatening muscle-wasting condition [252]. Intramuscular injection of a rAAV combined with a synthetic muscle promoter (Spc5.12) and microdystrophins (MD) has led to sustained levels of MD expression in a canine model of DMD [253]. Systemic and intramuscular delivery of rAAV-Spc5.12-MD vectors in mice shows a great amount of MD expression, distinct resistance to eccentric contraction and the absolute rescue of muscle mass [254,255,256]. Conversely, the majority of published studies do not report functional enhancements after treatment [257,258,259]. Dickson and colleagues demonstrate the effectiveness of locoregional and systemic delivery of rAAV2/8-Spc12, expressing a canine microdystrophin (cMD1) in restoring dystrophin expression in golden retriever muscular dystrophy (GRMD) dogs [260]. Locoregional administration exhibits a remarkable improvement of functional and histological factors, as well as a great amount of microdystrophin expression in limb musculature. Systemic delivery induces sustained levels of microdystrophin in skeletal muscles for almost 2 years after injection. Vector administration reveals no adverse immune consequences and toxicity in dogs. Therefore, this gene therapy vector has great potential for innovative therapies in DMD patients. AAV gene therapy allows transgene expression in cardiac myofibroblasts, which offers a great potential for heart regeneration after myocardial infarction (MI). AAV gene therapies have been reported to have transduced cardiomyocytes that are vital for cardiac function in mice after systemic injection [261,262]. In vivo systematic administration of AAV-6 and -9-mediated periostin promoter in the mouse heart targets gene expression to a myofibroblast-like lineage after MI [263]. AAV9 indicates a level of reporter gene expression higher than that of AAV6. This vector is capable of improving cardiac regeneration process by targeting gene expression to a large population of myofibroblast-like cells. Cardiac regeneration can also be achieved through the adenovirus carrying complementary DNA encoding cyclin A2 (Ccna2, a regulatory gene usually silenced after birth in the mammalian heart). Shapiro and co-workers administrated a null adenovirus and an adenovirus carrying cyclin A2 into a swine model with IM [264]. Six weeks after injection, cardiac contractile function data using magnetic resonance imaging (MRI) showed almost a 4% reduction in control pigs and about an 18% growth in ejection fraction of Ccna2-treated pigs. In vivo results indicate an enhanced cardiomyocyte number and cardiomyocyte mitoses, but decreased fibrosis in the pigs. Microscopy confirms that Ccna2 causes cardiac regeneration via cytokinesis of adult porcine cardiomyocytes.

#### 3.1.5. Nerve Regeneration

As a highly complex part of the body, the nervous system is responsible for behaviour, movement, and function. It can be classified into the central nervous system (CNS) and the peripheral nervous system (PNS). It is well known that the adult mammalian CNS cannot regenerate, which can cause severe health issues in a patient following a stroke, spinal cord injury, or traumatic brain. The failure of axonal regrowth in the mature mammalian CNS is related to an imbalance between intrinsic (the loss of axon regeneration capacity) and extrinsic (the inhibitory environment) factors [265]. The inability of axon regeneration and neurodegeneration leads to the loss of function of affected neuronal populations, which is a leading medical problem in neurobiology. Despite the regenerative failure of adult mammalian CNS, functional axon recovery can often occur after damage in the PNS owing to the self-repair and reactivate intrinsic growth capability of peripheral neurons following injury. Nonetheless, intense damages to peripheral nerves can also induce neurological deficits such as chronic pain or failure of reinnervation [266]. Gene therapy can successfully address these obstacles and promote axon regeneration in the adult mammalian CNS and PNS. The delivery of genes encoding neurotrophic factors, such as ciliary neurotrophic factor (CNTF) [267], brain-derived neurotrophic factor (BDNF) [268], and glial cell line-derived neurotrophic factor (GDNF) [269], is a powerful approach to reach axon regeneration. However, recent studies demonstrate that uncontrolled delivery of neurotrophic factors (specifically GDNF) prevents regeneration [269,270]. Thus, controlling the time window of neurotrophic factor delivery is a key step to deal with this problem, while achieving the long-term regulation of a therapeutic gene is still challenging in preclinical models [271,272]. Eggers et al. address this obstacle by employing a lentiviral immune-evasive doxycycline inducible GDNF vector (dox-i-GDNF) with the time-restricted GDNF expression in a longitudinal spinal cord lesion [273]. The lentiviral system enables an effective regulation of GDNF for almost 30 days and enhances long-term motor neuron survival (24 weeks), and axon regeneration by 8 weeks. Although continuous doxycycline-mediated delivery of GDNF improves motor neuron survival, it leads to axon entrapment (Figure 2). The use of knocked-out mice and optic nerve injury indicates long distance nerve regeneration. In recent research, the efficiency of AAV, RNA interference (RNAi), and pharmacological approaches to promote axon regrowth and target reconnection in wild-type mice was investigated in the presence of an optic nerve injury model [274]. The AAV expressing short hairpin RNA (shRNA) against the phosphatase and tensin homolog (shPTEN) improves retinal ganglion cell axon regeneration after crush injury. Nonetheless, the regeneration of AAV-shRNA was lower than PTEN KO mice models. This could be related to the incomplete gene silencing inherent of RNAi. The combination of AAV-shPTEN with AAV encoding CNTF and a cyclic adenosine monophosphate analogue provides long distance regeneration and target reconnection in mice.

An innovative and effective method for brain regeneration after ischemic injury is astrocyte-to neuron conversion. Although a broad range of studies has successfully converted glial cells into neurons, the total number of new neurons is unclear [275,276,277]. Utilisation of NeuroD1-based AAV gene therapy can efficiently regenerate approximately 35% of the total lost neurons in rodent animals and simultaneously preserve almost another 35% of both motor and cognitive deficits, showing remarkable recovery after brain injury [278]. Cre-FLEX system, along with an engineered AAV, were used to ectopically express NeuroD1 in reactive astrocytes. AAV was employed to affect both dividing and non-dividing astrocytes to enhance the number of neurons converted from astrocytes. Neuronal regrowth after cell conversion at both the protein and mRNA levels is confirmed by immunostaining and RNA sequencing. The astrocyte-converted neurons transferred long-range axonal projections to the target regions in a time-dependent manner. The cell conversion also resulted in a notable enhancement of motor and cognitive functions.

#### 3.1.6. Tooth Regeneration

Gene therapy approaches for tooth regeneration typically involves the regrowth of tooth, root, and periodontium (tissues surrounding or supporting teeth). Although gene therapy-based tooth regeneration has not yet reached its full potential, it is capable of delivering various genes and therapeutic factors to the tissue of interest that may mimic the complex natural process when compared with other therapies. The majority of current tissue engineering techniques have focused on the regeneration of periodontal tissues that contain alveolar bone, periodontal ligament, gingiva, and cementum. One clear explanation is that periodontal disease (also known as gum disease), a chronic gum inflammatory condition, is regarded as a leading cause of tooth loss in large number of adults [279]. Thus, it is desirable for gene therapy methods to recover the form and function of lost periodontium, resulting in partial or complete regeneration of teeth. Gene therapy approaches for periodontal regeneration have mostly concerned the use of platelet-derived growth factor (PDGF) [280]. Adenovirus encoding PDGF is reported to sufficiently transduce cementoblasts, gingival fibroblasts, osteoblasts, and periodontal ligament cells [281,282]. Effectiveness in preclinical periodontal defect models has been shown with adenovirus-encoding PDGF-B (PDGF factor B) to stimulate the regeneration of periodontal attachment [283,284]. Recently, Elangovan and colleagues used a non-viral gene therapy-based GAM approach to deliver polyplexes of polyethylenimine (PEI)-plasmid DNA encoding PDGF for periodontal regeneration in a rodent model [198]. The nonviral PEI-pPDGF-B vector shows efficient transfection of hGF and hPLF cells for the sustained generation of PDGF-BB. When compared with collagen alone or rhPDGF-BB protein, both PEI and nonviral PEI-pPDGF-B vector exhibit notably lower bone generation, which could be related to the dosing profile of PDGF-BB protein. Sustained PDGF gene delivery into the region lateral to the defect leads to a markedly higher number of inflammatory cells. Continuous PDGF-BB production through non-viral gene therapy delays bone healing via prolonging inflammatory response. Another interesting strategy for periodontal regeneration is the combination of gene therapy with autologous stem cell. Ex vivo viral gene transfer using stem cells to express the BMP-2 gene in rabbits allows cementum regeneration with Sharpey’s fibre insertion and higher quantity of osteoprogenitor cells [285]. Such vectors result in a safer relationship between the components of the periodontal attachment than the direct the usage of BMP-2. When combined with canine periodontal ligament stem cells, adenoviral BMP-2 generates mature thick lamellar bone in beagle dogs with peri-implantitis defects [286]. This vector also shows a considerably higher formation of new bones around pre-implantitis defects. Electroporation gene delivery has been studied as an efficient vector for tooth regeneration. For example, Akamine and co-workers investigated the gene transfer of growth/differentiation factor 11 (Gdf11) to pulp cells for tissue regeneration of pulp and dentin repair [287]. The Gdf11 cDNA plasmid induced the expression of dentin sialoprotein in mouse dental papilla via electroporation. Gdf11 gene transfer to pulp cells in an amputated tooth in vivo allows the reparative dentin generation.

#### 3.1.7. Hair Cell Regeneration for Hearing Loss Treatment

The mammalian cochlea hair cells (HCs) consist of inner hair cells (IHCs) and outer hair cells (OHCs) that are responsible for sound detection and processing [288]. They are surrounded by a heterogeneous group of cells known as supporting cells (SCs), which are responsible to protect HC and maintain its environment. Inner ear gene therapy holds excellent potential to promote HC damage [289]. Viral gene therapies have been extensively explored for in vivo inner ear studies of hereditary hearing loss to promote auditory function [290,291,292,293]. Most of these studies employ AAV gene therapies. Regardless of the infection efficiency of traditional AAVs for cochlear IHCs, the efficient infection of gene therapies is reported to be low for both outer hair cells [290,294] and SCs [295,296]. Thus, the desirable inner ear gene therapy should be capable of promoting higher infection efficiency to achieve sufficient hearing recovery. In recent research, Isgrig et al. evaluated the infection efficiency of gene therapy for inner ear by using synthetic viral vectors, namely AAV2.7m8 and AAV8BP2, in a mouse model [297]. The results revealed that AAV2.7m8 cannot only effectively affect both inner and outer hair cells, but it can also sufficiently affect the subdivision of SCs (inner phalangeal cells and inner pillar cells) that are believed to improve hair cell regeneration. Although adult mammalian hair cells are unable to regenerate after damage, the regeneration of supporting cells is a useful approach for hair cells regeneration. However, the AAV2.7m8 vector does not target other types of supporting cells, including Hensen’s cells, Deiters cells, and outer pillar cells. To address this issue, Zhong and colleagues developed an AAV-inner ear (AAV-ie) for gene therapy in a mouse inner ear [298]. Approximately 100% of supporting cells in the utricle and all the cochlear supporting cell types in the cochlea were efficiently targeted by AAV-ie, showing the suitability of this viral vector for hair cell regeneration and correcting genetic defects in supporting cells. The AAV-ie-mediated Atoh1 expression, an essential gene for the formation of ear hair cells, led to the generation of a large number of new hair cells in the sensory region (Figure 3). Despite the potent regeneration ability of AAV-ie, the newly formed hair cells failed to repair mature hair cells (Figure 3d–i). Consequently, future assessments are definitely required to achieve complete hair cells regeneration.

#### 3.1.8. Skin Regeneration

Skin, the largest organ of the body, includes epidermis, dermis, and skin appendages (e.g., sebaceous gland, hair follicle), which acts as a potent shield against various external factors. It is regarded as an ideal tissue for gene therapy approaches, because skin is effortlessly available tissue related to monogenetic and chronic diseases [299]. The non-viral gene transfer of keratinocyte growth factor (KGF) and IGF-I via liposomal cDNA complexes indicates dermal and epidermal regeneration in rat models [300]. Specifically, non-viral cDNA gene transfers show dose-dependent behaviours with the highest repair efficiencies at a concentration of 2.2 μg for both growth factors. The KGF, IGF-I, and their gene vector individually enhance KGF, IGF-I, FGF, VEGF, and collagen type IV expression, as well as increased re-epithelisation [301]. In contrast, there was no effect on collagen type I and III expression. When compared with other groups, liposomal transfer of KGF and IGF-I cDNA improves dermal and epidermal regeneration in rats through increased neovascularisation and VEGF expression. The efficiency of non-viral liposomal gene therapy for dermal and epidermal regeneration is also reported in preclinical studies using pig models [302]. On days 2 and 4, improved PDGF-mRNA and protein expression were observed, while 9 days post-burn increased wound re-epithelisation and graft adhesion. The non-viral transduction of VEGF to epidermal stem cells is also an attractive strategy for skin regeneration. The vector was prepared via linking β-cyclodextrin with polyethylenimines, and it was used in combination with a 3D gelatin-based scaffold for the transfection and culture of epidermal stem cells [303]. In comparison with two-dimensional plates, the gene transfer-based 3D system led to prolonged VEGF expression with a higher amount at day 7 in epidermal stem cells. Skin re-epithelisation, dermal collagen fabrication, and hair follicle reformation were also present. The non-viral gene therapy systems adjusted the distribution of different types of collagen, leading to inhibition of scar formation. Another alternative approach involves chemokines-based gene therapy in exogenous bone-marrow-derived MSCs for skin tissue regeneration. MSC therapies in wound tissue regeneration suffer from a limited number of transplanted cells at the wound area. One promising solution involves the use of chemokines that is thought to be a mediator of neo-vascularisation, re-epithelialisation, inflammation, and regeneration process in skin repair [304,305,306]. Higher expression of chemokine ligand CXCL16 in a wound site and its cognate receptor (CXCR6) on MSCs may provide effective treatments. However, this strategy alone is not effective, because it does not improve MSC recruitment at the wound bed. Gene therapy can successfully address these issues by transplanting Cxcr6-overexpressing MSCs towards CXCL16 at the wound area, causing higher directional migration, endothelialisation, and epithelialisation [307]. CXCR6 gene therapy combined with MSC resulted in the ameliorated recruitment and engraftment at the wound site in both type I and type II (db/db) diabetic, as well as non-diabetic mice models. It has led to improved collagen deposition, neo-vascularisation, and re-epithelialisation regenerated skin (Figure 4).

#### 3.1.9. Disc Regeneration

Many people suffer from lower back pain thought to be caused mainly by disc degeneration [308]. The degeneration process generally starts from the inner nucleus pulposus with a loss of proteoglycans and cell numbers [309]. To treat disc degeneration by the regeneration of nucleus pulposus, Feng et al. developed a non-viral cationic block copolymer gene delivery vector based on the combination of pDNA, PEG-block-poly{N-[N-(2-aminoethyl)-2-aminoehtyl]aspartamide} [PEG-b-PAsp(DET)] and poly(N-isopropylacrylamide)-block-PAsp(DET) [PNIPAM-b-PAsp(DET)] at room temperature in rat tail discs [310]. The mixed polyplex micelles with heterogeneous hydrophobic/hydrophilic coronas were obtained by increasing the temperature from 25 °C to 37 °C. When compared with the basic polyplex micelles, the gene transfection efficiency of micelles in nucleus pulposus cells was notably greater, which can be explained by the synergistic result of increased colloidal stability as well as low cytotoxicity. The high expression of heme oxygenase-1 (HO-1) in nucleus pulposus cells transfected by micelles loading HO-1 pDNA enhanced the phenotype-associated genes (SOX-9, type II collagen, aggrecan), and meanwhile, reduced the expression of COX-2 and matrix metalloproteinases 3 (MMP-3) induced by interleukin-1b (IL-1b). In addition, the injection of micelles loading HO-1 pDNA improved glycosaminoglycan content, and showed lower inflammatory responses than that of blank micelles. Self-assembled polyplexes consisted of hyperbranched polymer (HP) with high pDNA binding affinity attempts to sufficiently transfect pDNA into nucleus pulposus cells [311]. These polyplexes with double-shell structures were encapsulated in biodegradable nanospheres (NS), not only allowing temporally controlled release of pDNA-delivering polyplexes, but also providing suitable transfer of pDNA into cells. The nanofibrous spongy microspheres (NF-SMS) were injected together with NS for the delivery of pDNA encoding orphan nuclear receptor 4A1 (NR4A1). A rat-tail degeneration model indicated a reduction in the pathogenic fibrosis of nucleus pulposus tissue through the NR4A1 pDNA. Eventually, the combined delivery of NF-SMS and the NR4A1 pDNA delivery NS enhanced the regeneration of nucleus pulposus (Figure 5).

### 3.2. Biomaterials-Mediated Regeneration

The development of materials capable of eliminating regulatory approaches, regulating cell behaviour, and reducing the need for in vitro culture is crucial for the design of gene therapy-based tissue regenerations. Either natural or synthetic biomaterials can be classified into hydrogels or scaffolds. From the in vivo prospective, biomaterials should mimic the physical structure, chemical composition, and biological function of native ECM. These materials should be biocompatible and promote the formation of new tissues. This section describes the different classes of biomaterials that have been applied in regenerative medicine through viral or non-viral gene therapy procedures.

#### 3.2.1. Hydrogels-Mediated Regeneration

Hydrogels are natural or synthetic crosslinked polymers capable of absorbing large amounts of water. Due to their adjustable physical, chemical, and biological properties, inherent similarity to the natural ECM and living tissues, as well as excellent biocompatibility, hydrogels have been extensively explored in tissue regeneration areas [312]. Collagen and hyaluronic acid (HA) hydrogels have been clinically employed for cartilage repair [313]. The lower mechanical ability of collagen hydrogels than cartilage, and due to the fact that chondrocytes are naturally attached to HA-based matrices, all make HA hydrogels better candidates for cartilage repair [314]. The attachment of chondrocytes to HA hydrogels preserves the differentiated chondrocyte phenotype and stimulates ECM formation [315]. Yet, their limited half-life upon exposure to an inflammatory area and low mechanical stabilities can limit their applications [316]. Encapsulating viral and non-viral gene delivery systems inside hydrogels prevents unwanted loss during the transit to the desired area, allows controlled delivery at the target tissue, and enables suitable localised gene therapy with lower nonspecific spreading to other tissues [317]. Consequently, hydrogel efficiently enhances the local retention time of the gene vector at the tissue of interest, and improves the internalisation of these vectors by the target tissue, as that is the main problem in non-viral gene therapies-based nanoparticles [318]. The DNA uptake of alginate hydrogel is improved by the DNA-PEI aggregates [319]. Incorporation of PEI-functionalised GO nanosheets (fGO) complexed with DNA within methacrylated gelatin hydrogel-based non-viral vector allows controlled and localised gene therapy in a rat model with acute myocardial infarction [317]. The administration of hydrogel-based gene therapy in the peri-infarct regions considerably promotes myocardial capillary density and decreases the scar site. The efficiency of hydrogel is also reported in cartilage repair applications in which a poloxamer-poloxamine hydrogel delivered a rAAV-TGFB1 vector improves Type II collagen and proteoglycan in osteochondral defects [320]. Similarly, a fibrin hydrogel carrying a rAAV-TGFB1 vector enhances expression of cartilage-specific genes in MSCs [321]. In a recent work, a gene-hydrogel microenvironment consisting of PEG hydrogel loaded agomir was used to adjust the fabrication/catabolism balance of the ECM in the nucleus pulposus to promote disc regeneration in a rat [322]. As a cholesterol-, methylation-, and phosphorothioate-modified synthetic microRNAs (miRNAs), agomir is capable of mimicking the role of miRNA to adjust the expression of the target gene [323]. An injectable PEG hydrogel with antimicrobial, degradable, self-healing, and superabsorbent properties was produced via Ag-S coordination of 4-arm PEG-SH and silver ion solution. The mechanical stability of the PEG-hydrogel was finely suited to the condition of disc. Meanwhile, Agomir874 was fabricated to down regulate the expression of MMPs in nucleus pulposus. The resultant hydrogel formed a gene-hydrogel microenvironment in situ in an intervertebral disc that decreased the expression of MMPs, adjusted the synthesis/catabolism balance of ECM in the nucleus pulposus of rat disc, and promoted tissue regeneration [322] (Figure 6).

#### 3.2.2. Scaffolds-Mediated Regeneration

The development of scaffolds-based biomaterials in regenerative medicine applications requires the consideration of scaffold material, size, shape, and their encapsulation abilities for therapeutic agents. Scaffolds should be biocompatible with suitable mechanical stabilities, and provide gene therapy to the sites of interest, leading to tissue repairs or generation of new tissues. A 3D-woven poly(e-caprolactone) (PCL) scaffold-mediated transduction of hMSCs with lentiviral vectors carrying TGF-β3 shows excellent differentiation for cartilage-like ECM in chondrogenic cultures [324]. This strategy can facilitate the extensive cell manipulation ex vivo by using in situ tissue engineering. Using solid freeform fabrication along with biomineral coating on Poly (L-lactic acid) (PLLA) and PCL scaffolds enhance bone formation in mice [325]. The mineralised tissue formation was only observed in scaffolds with BMP-7-transduced human gingival fibroblasts. At 3 weeks, the bone ingrowth between the coated and uncoated scaffolds was almost similar, while coated scaffolds exhibited incredibly higher bone ingrowth than that of uncoated scaffolds at 10 weeks. Due to improved bone ingrowth, coated PLLA scaffolds demonstrated higher mechanical stabilities. Another substantial factor in regenerative ability of scaffold-mediated gene therapies is spatiotemporal control of gene expression that can conversely affect vector dispersion. One effective solution is the use of the chemical vapor deposition (CVD) method for targeting the effects of gene therapy to the tissue of interest [326]. Recently, Pilipchuk et al. developed micropatterned scaffold-based growth factor gene delivery to regenerate periodontal defects in athymic rats (Figure 7) [327]. Amorphous PCL scaffolds were fabricated for bone regeneration and polylactic-co-glycolic acid (PLGA)/PCL scaffolds were constructed for ligament repair. Using the CVD method and adenoviral vectors, the former scaffolds were encapsulated with BMP-7, and the latter with PDGF-BB. The combined growth factor gene delivery with surface topography revealed better tissue regeneration than other treatments alone. Apart from the above-mentioned examples that mainly concern viral vector therapies in regenerative medicine, non-viral gene therapies can also be applied in the regeneration of tissues. For example, O’Brien and colleagues report the efficacy of PEI and collagen-based scaffolds for bone regenerations through non-viral GAM technology [328]. When seeded with rMSCs, PEI polyplex GAMs, synthesised from 2 μg of pDNA per scaffold, can achieve sustained release of gene expression for 21 days. The most prolonged levels of transgene expression were observed for the collagen–nHa GAM (two weeks). Effectiveness was subsequently obtained with non-viral vectors carrying PDGF-B [199], VEGF and BMP2 [329], and BMP-2 [330] for bone regenerations.

### 3.3. Nanomaterials-Mediated Regeneration

The clinical application of nanomaterial-based gene therapy is restricted to cellular internalisation, aggregation in physiological fluids, and lack of targeting ability, endosomal escape, biodegradation, and biocompatibility [75]. Some of these challenges are satisfactorily addressed specifically in non-viral gene therapies (See Section 2.2.1). Biocompatibility of nanoparticle-based gene therapy plays a vital role in clinical applications. Natural polymers (e.g., cyclodextrin, chitosan) and polypeptide-based cationic polymers are potent candidates for gene therapies that have reached clinical applications mainly due to their preferable biocompatibilities [331]. Lipid nanoparticles can also serve as promising carriers for gene delivery with the advantages of behaviour similar to the lipidic membranes and high biocompatibility, which leads to easier cell penetrations [332]. Other examples of biocompatible nanomaterials-guided gene therapy are PEI [333], polyamidoamine [334], poly(β-amino ester) [335], and disulfide cross-linked polymers [336]. Nanomaterial gene therapies in regenerative medicine applications can also be hampered by aggregation and nonspecific adsorptions as the result of extensive positive charges on the surface of cationic nanomaterials. Even neutral particles face fast clearance in the absence PEG by forming large aggregates and being adsorbed via serum albumin [337]. One suitable method to enhance accumulation in targeted tissues and reduce nonspecific interactions with serum proteins in the bloodstream is PEGylation [338]. Polyanions is another facile strategy that reduces nonspecific interactions, stabilises the nanoparticles, and extends blood circulation [339]. Graphene oxide (GO) is an effective gene carrier that does not cause toxicity at a dosage less than 100 mg/kg [340] and shows in vivo biocompatibility in mice even after 3 months [341]. GO can serve as a natural antioxidant to reduce inflammatory polarisation of macrophages (M1) through the reduction of reactive oxygen species (ROS). It can also sufficiently deliver interleukin-4 plasmid DNA (IL-4 pDNA) for myocardial infarction therapy [342]. The tissue regeneration phase can be achieved by timely termination of an inflammatory phase. Thus, a macrophage-targeting/polarizing GO complex (MGC) was developed via modification of GO with PEI and folic acid−PEG (Figure 8A). MGC decreased ROS in immune-stimulated macrophages and generated lower levels of inflammatory biomarkers (Figure 8B). DNA-functionalised MGC polarised M1 to M2 macrophages and improved the expression of reparative biomarkers required for heart regeneration. The administration of MGC/IL-4 pDNA in mice models decreased inflammation and cardiac fibrosis, higher blood vessel density, as well as enhanced heart function. To promote bone regeneration via silencing SOST genes with a specific siRNA, mesoporous silica nanoparticles were modified with PEI [343]. The modified nanoparticles led to the effective delivery of SOST siRNA and osteostatin (an osteogenic peptide) inside cells. In vitro results exhibit great amounts of the osteogenic markers, and a high capacity to silence SOST. The co-injection of SOST siRNA and osteostatin in the femoral bone marrow of mice improves the expression of early markers of osteogenic differentiation that may promote bone formation. Previous studies highlight the successful transfection capability of nanoplexes delivering BMP-2 and FGF-2 in vitro [344]. It has been demonstrated that the combination of these gene allows the synergistic effect on osteogenesis. Non-viral gene delivery of FGF-2 and BMP-2 can synergistically improve bone regeneration in a diabetic rabbit model [345]. These nanoplexes were synthesised by mixing plasmid DNA encoding BMP-2 or FGF-2 in PEI complexes. In comparison with individual delivery of nanoplexes in diaphyseal long bone radial defects (either BMP-2 or FGF-2), the PEI (pBMP-2 and pFGF-2) indicated higher bone mineral density and higher formation of new bone in vivo. The mixture of PEI (pBMP-2 and pFGF-2) with collagen scaffold notably improves bone regeneration.

## 4. Recent Progress in Non-Viral Gene Therapy

The m-RNA vaccination against coronavirus demonstrated the safety and efficiency of non-viral delivery systems for nucleic acid-based agents [346,347,348]. The lipid composition is not trivial and contains synthetic molecules, such as PEG-lipids and lipid-amine molecules. The Pfizer/Biontech vaccines were delivered within lipid nanoparticles that contained 4-hydroxybutyl)azanediyl)bis(hexane-6,1-diyl)bis(2-hexyldecanoate), 2 [(polyethylene glycol)-2000]-N,N-ditetradecylacetamide, 1,2-Distearoyl-sn-glycero-3-phosphocholine, and cholesterol. The Moderna vaccine contained the following lipids: Octanoic acid, 8-[(2-hydroxyethyl)[6-oxo-6-(undecyloxy)hexyl]amino]-, 1-octylnonyl ester (SM-102), polyethylene glycol [PEG] 2000 dimyristoyl glycerol [DMG], cholesterol, and 1,2-distearoyl-sn-glycero-3-phosphocholine [DSPC]), tromethamine, tromethamine hydrochloride, acetic acid, sodium acetate trihydrate, and sucrose. [349].

However, these carriers are useful for vaccination where a short time of activity is required to generate enough spikes that induces the immune system to generate immunisation. For non-vaccination therapies such as cancer treatment, nucleic acid-based medications should be delivered for weeks at a relatively high dose. For example, for the treatment of pancreatic cancer, a PLGA implant loaded with 4 mg of siRNA is inserted into the tumorous tissue that releases its siRNA content for a period of several months [350].

## 5. Conclusions

Regenerative medicine based on gene therapy technology is very promising by developing safe, rapid, and cheap polymeric materials in order to enhance tissue regeneration by the localisation of DNA encoded specific protein. Recent development in the synthetic materials and their advantages in design of suitable vehicles for DNA delivery to cells and tissues are reviewed in this paper. Although many new technologies have been developed and successfully applied for stable gene transection in vivo, a lot more research is needed to be developed and applied for regenerative medicine therapy.

## Figures and Tables

**Figure 1 pharmaceutics-15-00856-f001:**
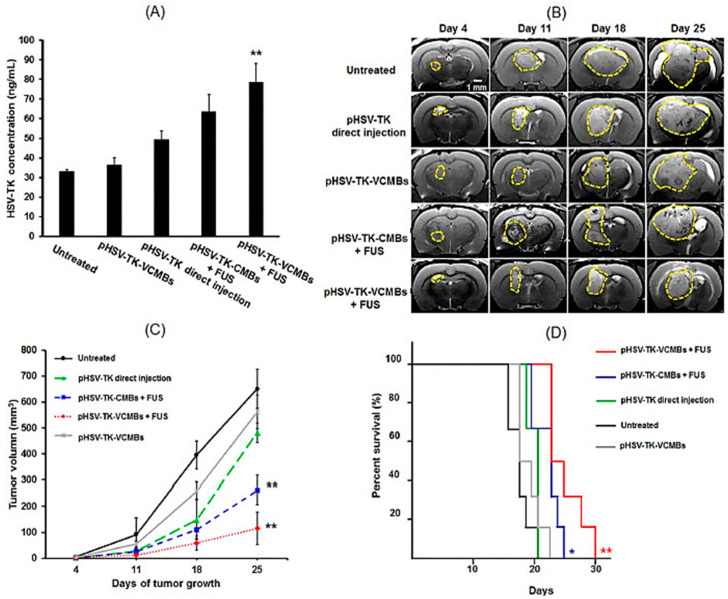
Anti-tumour efficiency of pHSV-TK-VCMBs with FUS. (**A**) Quantitative results of HSV-TK expression after using pHSV-TK-VCMBs and FUS treatment (N = 4 per group, compared with untreated group). (**B**) T2-weighted MRI images of brain tumours at 4, 11, 18, and 25 days after tumour implantation. Rats received no treatment (control group) or treatment on day 7 with: direct injection, pHSV-TK-CMBs, or pHSV-TK-VCMBs. (**C**) Tumour volume assessed by MRI imaging (N = 6 per group, compared with untreated group). (**D**) Analysis of animal survival rate (N = 6 per group, compared with untreated group) (* *p* < 0.5, ** *p* < 0.01). Reprinted with permission from Ref. [165], Elsevier, copyright 2014.

**Figure 2 pharmaceutics-15-00856-f002:**
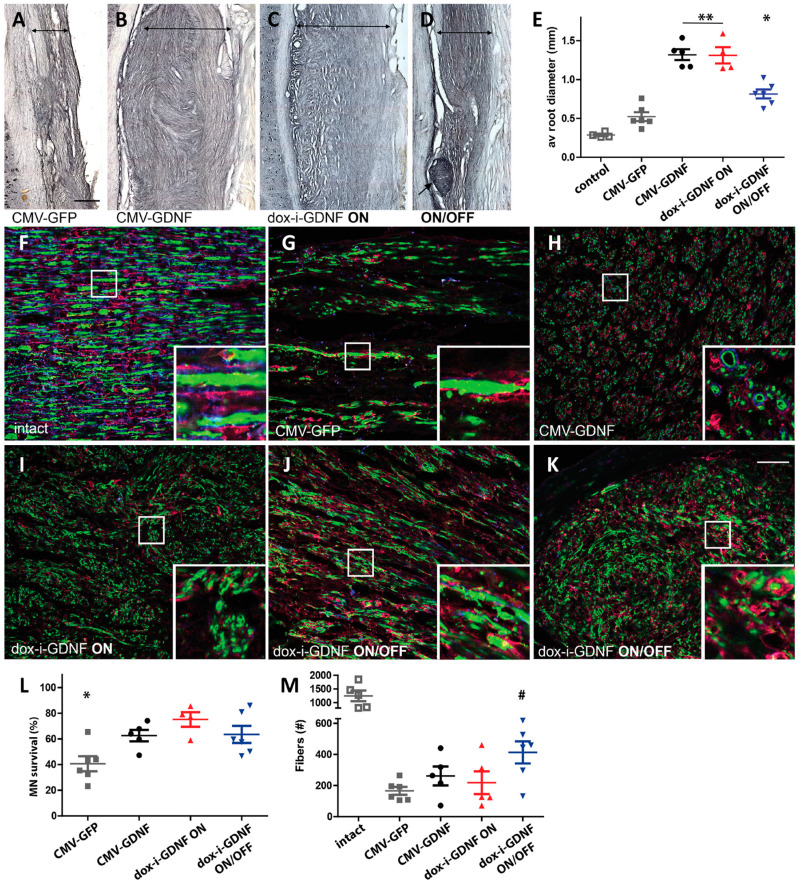
Timed GDNF protein expression reduces axonal coil formation, improves distal axon outgrowth, and promotes motor neuron survival—results of Experiment 1. (**A**–**E**) The diameter of ventral roots adjacent to the spinal cord was analysed in ChAT-stained longitudinal sections 12 weeks post-reimplantation. Compared to intact ventral roots, the average diameter increases slightly in CMV-GFP the control group (*p* < 0.005). (**B**,**C**,**E**) Continuous GDNF protein in the CMV-GDNF and dox-i-GDNF ON groups results in a significant increase in the ventral root diameter. Exposure to time-restricted GDNF expression results in a significant size reduction (**D**,**E**), though small isolated nerve coils are occasionally found. ((**D**) arrow). Constitutive GDNF expression results in densely packed thin fibres growing in a swerving fashion as shown in high magnification ventral root images taken 2 to 8 mm from the implantation site ((**H**,**I**) MBP; blue, Neurofilament; green, S100; red). (**J**) In the ON/OFF group ventral roots were filled with axon profiles displaying a more longitudinal growth pattern. (**K**) However, coils observed in the ON/OFF group inside the small nerve (shown in (**D**)), and the axon growth orientation was disrupted similar to the groups with constitutive GDNF expression. (**L**) Motor neuron survival at 12 weeks improved in all GDNF treated groups. Long distance motor axon outgrowth, quantified in transverse sciatic nerve sections stained for ChAT (**M**), was significantly increased in the ON/OFF animals. (**E**) * *p* < 0.002 versus CMV-GDNF and ON, ** *p* < 0.0001 versus CMV-GFP. (**L**) * *p* < 0.01. (**M**) # *p* < 0.008 versus CMV-GFP, *p* < 0.05 versus CMV-GDNF and ON). Scale bar in A (**A**–**D**) = 250 µm, K (**F**–**K**) = 50 µm. Data represent individual animals and are expressed as mean ± SEM. Reprinted from ref. [273] with permission from Oxford University Press, copyright 2019.

**Figure 3 pharmaceutics-15-00856-f003:**
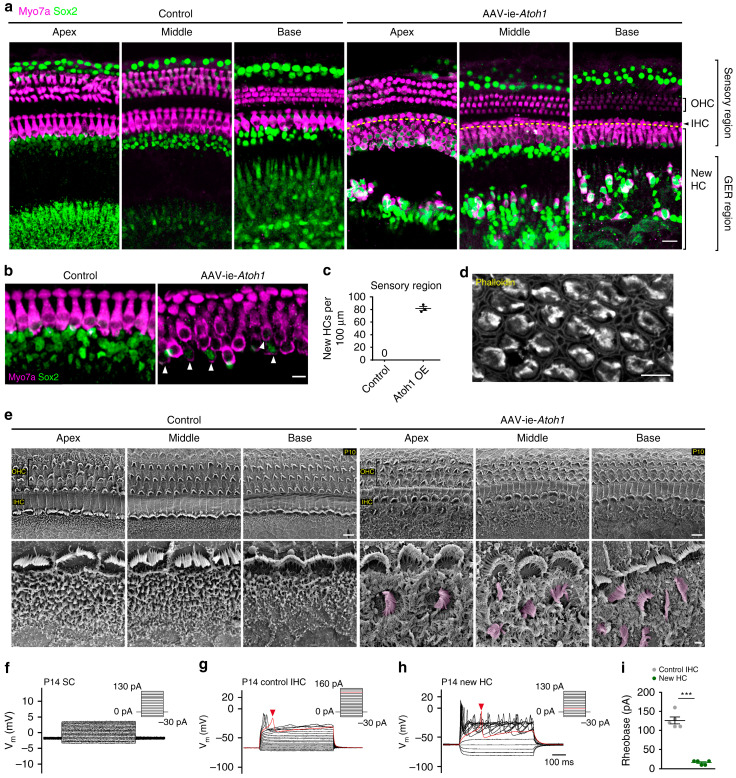
Adeno-associated virus-inner ear-Atoh1 (AAV-ie-Atoh1) induces new hair cells (HCs) in vivo with stereocilia. Mice were injected with AAV-ie-Atoh1 (1 × 10^10^ genome-containing (particles) (GCs)) at P0, and the cochlea was harvested at P14. (**a**) Representative confocal projection image of control and AAV-ie-Atoh1 cochlea. Scale bar, 20 μm. (**b**) Magnification of inner HC (IHC) region of control and AAV-ie-Atoh1 cochlea. White arrows indicate both Sox2-and Myo7a-positive new HCs. Green: Sox2; magenta: Myo7a. Scale bar, 10 μm. (**c**) Number of Myo7a-positive new HCs per 100 μm in sensory region. Data are shown as mean ± SEM. N = 3. Source data are provided as a Source Data file. (**d**) Representative confocal image of phalloidin staining of new HCs in AAV-ie-Atoh1 cochlea. Scale bar, 10 μm. (**e**) Scanning electron microscopy (SEM) images of AAV-ie- and AAV-ie-Atoh1-injected cochlea at apical, middle, and basal regions. Regenerated HC-like cells were artificially coloured magenta. Scale bars, 5 μm (upper panels), 1 μm (lower panels). (**f**) Representative membrane responses of P14 supporting cells (SCs) to current. The trace shows action potential generation in response to 10 pA injections. N = 5. (**g**) Same as (**f**), but for P14 IHCs. The trace shows action potential generation in response to 10 pA injections. The first action potential was generated by 130 pA injection (red arrow). (**h**) Same as (**f**), but for P14 new HC injected with AAV-ie-Atoh1 and the first action potential was generated by 20A injection (red arrow). (**i**) Average responses show significant difference between IHCs and regenerated new HCs. Data are shown as mean ± SEM. *p* Value is calculated by Student’s *t* test. *** *p* < 0.001. N = 5. Reprinted from Ref. [298] with permission from Nature, copyright 2019.

**Figure 4 pharmaceutics-15-00856-f004:**
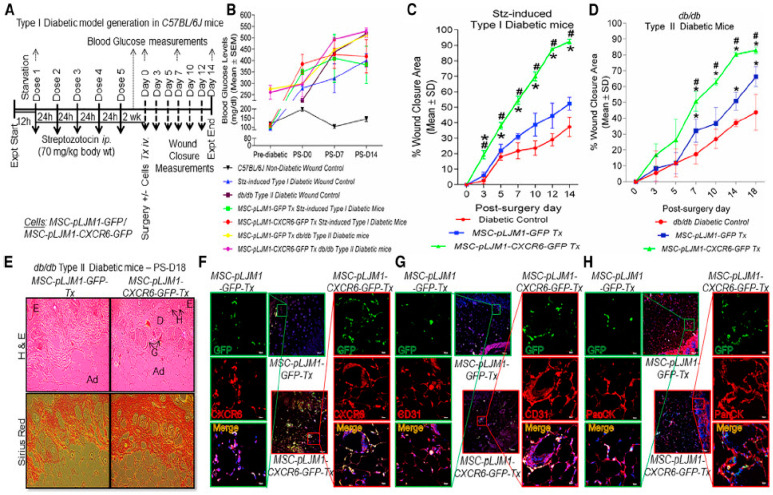
Transplantation of MSCs with CXCR6 gene therapy efficiently regenerated skin in Type I and II diabetic mice. (**A**) Flow chart representing Type I diabetes model generation. (**B**) High fasting blood glucose levels (>150 mg/dL) was observed in both Type I and II (db/db) diabetic mice. Higher percent wound closure area in diabetic mice transplanted with MSCs-Cxcr6. (**C**) Type I Stz-induced C57BL/6J and (**D**) Type II db/db mice. (**E**) Higher H&E and Sirius red staining in db/db mice transplanted with MSCs-Cxcr6 depicting an organised layer of dermis with the presence of glands and hair follicles in the regenerated wounds, as compared with control MSCs transplanted groups (E, epidermis; D, dermis; H, Hair follicles; G, sebaceous glands; Ad, adipose layer). Increased co-immunostaining of (**F**) GFP/CXCR6, (**G**) GFP/CD31, and (**H**) GFP/PanCK, suggesting more recruitment, engraftment, neo-vascularisation, and epithelialisation in db/db mice transplanted with MSCs-Cxcr6, as compared with MSC-control. (n = 3 replicates/wound, N = 4–5 mice/group; *p* < 0.05 as compared with * pLJM1-EGFP control; # diabetic control). Reprinted from Ref. [307] with permission from Elsevier, copyright 2020.

**Figure 5 pharmaceutics-15-00856-f005:**
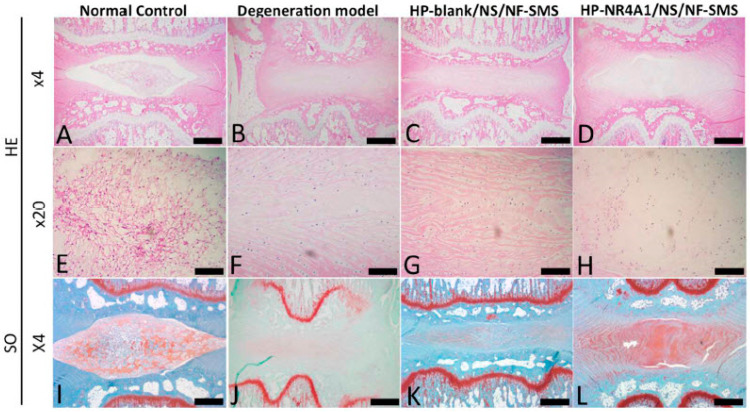
Hematoxylin and eosin (HE) and safranin-O (SO) stained disc images from different experimental groups at four weeks after injection. In the intact discs of the normal control group (**A**,**E**), the oval-shaped nucleus pulposus occupied a large volume of the disc space as viewed in the HE-stained midsagittal cross sections. The nucleus pulposus area was stained strongly with safranin-O (**I**), indicating a high glycosaminoglycan (GAG) content. In the degeneration model group (**B**,**F**,**J**), the disc space collapsed, with evident fibrous tissue invasion. Inhomogeneous fibrous tissue was found in the HP-blank/NS/NF-SMS group (**C**,**G**,**K**), and the nucleus pulposus area was stained almost negatively by safranin O. Although discs in the HPNR4A1 pDNA/NS/NF-SMS group (**D**,**H**,**L**) still displayed some degree of degeneration, therapeutic efficacy was obvious compared to the degeneration model group. The nucleus pulposus area in this group was more similar to the normal control group than other groups and was stained positively with safranin O. Scale bar: (**A**–**D**,**I**–**L**) 250 μm, (**E**–**H**) 100 μm. Reprinted from Ref. [311] with permission from Elsevier, copyright 2017.

**Figure 6 pharmaceutics-15-00856-f006:**
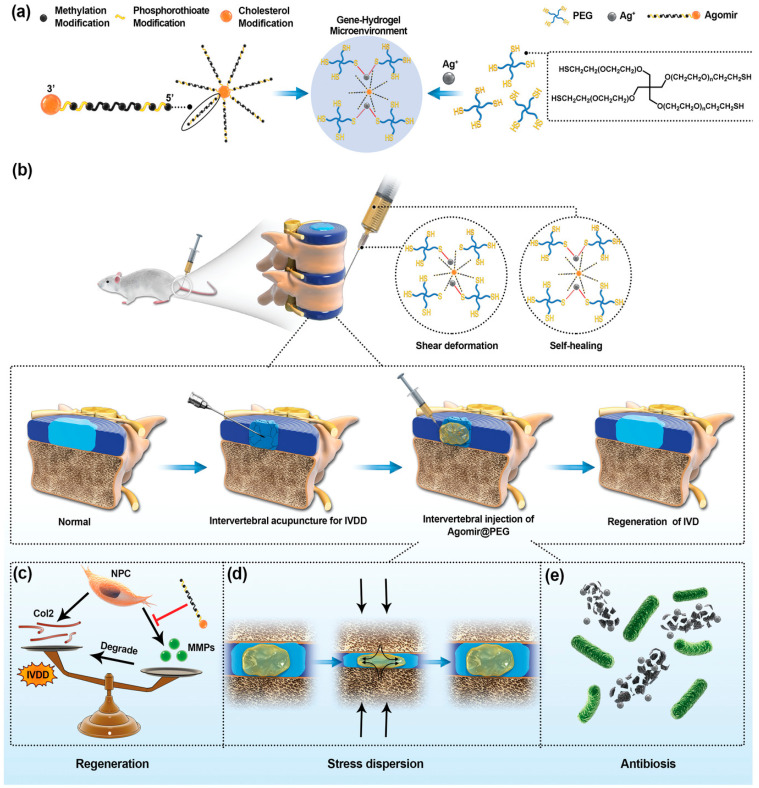
Gene-hydrogel microenvironment for regeneration of intervertebral disc degeneration (IVDD). (**a**) The construction of gene-hydrogel microenvironment. (**b**) Agomir@PEG was injected into the intervertebral space to construct the gene-hydrogel microenvironment. (**c**–**e**) The multi-functions provided by the gene-hydrogel microenvironment, matching the regeneration of IVDD. Reprinted from Ref. [322] with permission from WILEY-VCH Verlag GmbH & Co. KGaA, Weinheim, Germany, copyright 2020.

**Figure 7 pharmaceutics-15-00856-f007:**
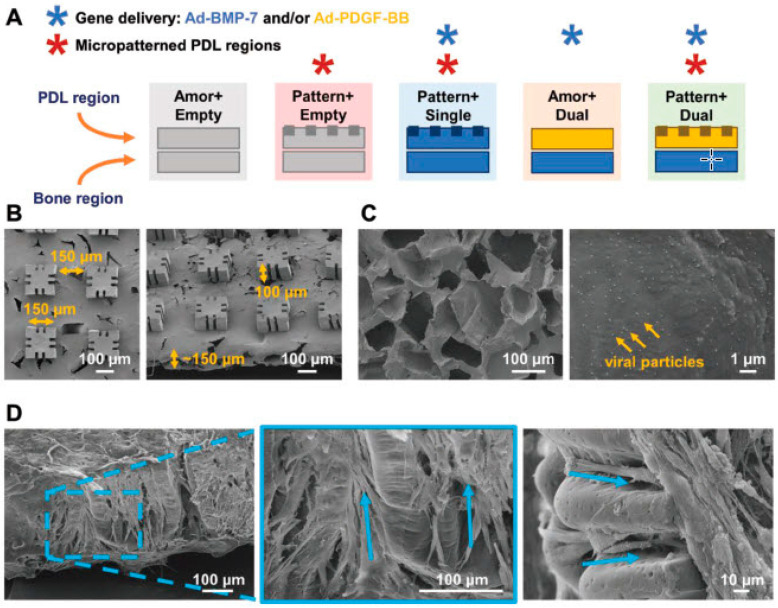
Scaffold design for periodontal tissue neogenesis. (**A**) Schematic representation of the polymeric scaffolds designed to deliver gene therapy vectors and/or micropatterned surface topography to bony defects. The periodontal ligament (PDL) regions were either patterned or amorphous, made of PLGA/PCL, and seeded with human periodontal ligament cells (hPDLs), except Pattern+Single, which received human gingival fibroblasts (hGFs). The bone regions were amorphous, made of PCL, and seeded with human gingival fibroblasts (hGFs). All regions were CVD-coated to immobilise adenoviral genes for Ad-BMP-7 (blue), Ad-PDGF-BB (yellow), or Ad-empty (gray). (**B**) SEM images of patterned PDL region, showing the pillar and groove dimensions. (**C**) SEM images of CVD-coated, PCL, porous base with immobilised adenoviral particles (10^12^ PN mL^−1^). (**D**) SEM images of hPDLs aligned with the micropatterning, 3 d after seeding. Blue arrows indicate the alignment of cells along the grooves embedded within the scaffold pillars, as well as in the interpillar regions. (Ad; adenovirus) Reprinted from Ref. [327] with permission from WILEY-VCH Verlag GmbH & Co. KGaA, Weinheim, copyright 2018.

**Figure 8 pharmaceutics-15-00856-f008:**
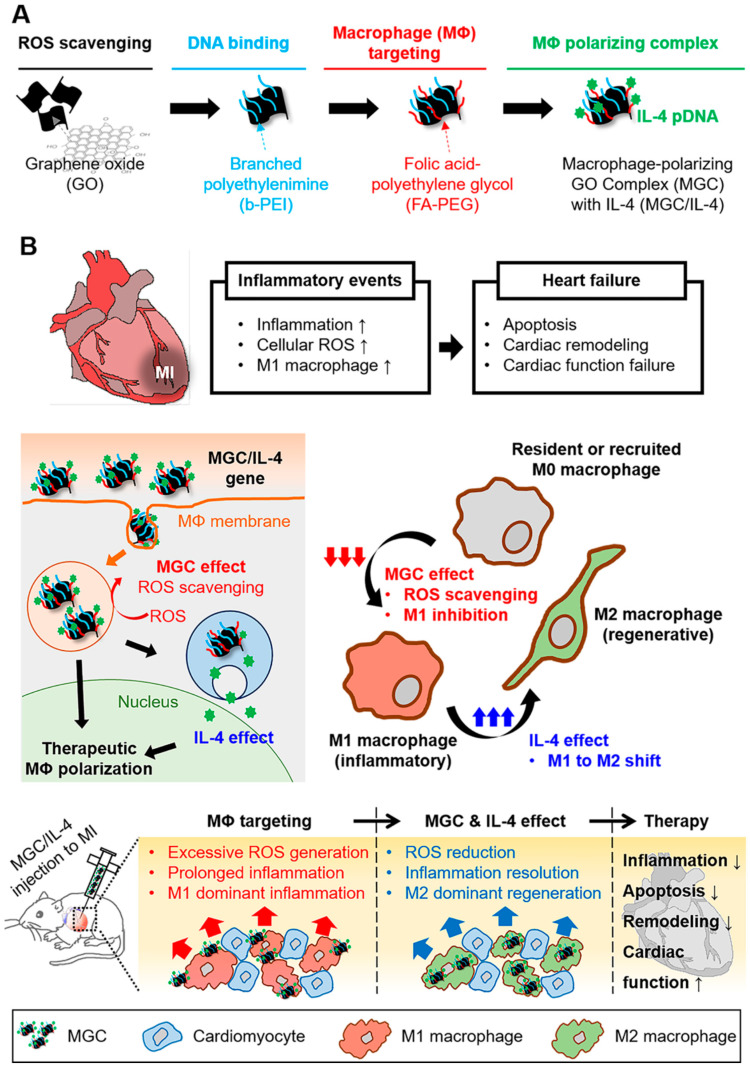
Schematic illustration of the preparation of the macrophage-polarizing GO complex (MGC)/IL-4, progression of heart failure after MI, and the therapeutic mechanisms of MGC/IL-4 pDNA to treat MI. (**A**) Stepwise preparation of MGC or MGC/IL-4 pDNA and the role of each chemical conjugation. (**B**) Progression of heart failure after MI and the therapeutic mechanisms of MGC/IL-4 pDNA in cardiac repair. Reprinted from Ref. [342] with permission from American Chemical Society, copyright 2018.

**Table 1 pharmaceutics-15-00856-t001:** Comparison between viral and non-viral vectors in gene therapy.

Gene Delivery Methods	Advantages	Disadvantages	Ref.
Viral-mediated methods	High transfection efficiency	Risk of viral infection Immunogenicity, toxicity limited size of gene, one copy Expensive, long procedure Stability and regulatory issues Limited to certain tissues and cells	[55]
Non-viral-mediated methods	No infection Low toxicity No limited on the size of gene Cheap, easy to prepare Stable Applicable to all tissues and cells	Low transfection efficiency	[58]

**Table 2 pharmaceutics-15-00856-t002:** Gene delivery strategies for ex vivo and in vivo bone regeneration.

Gene Delivery	Approach	Vector	Gene	Model	Remarks	Ref.
Viral	Ex vivo	Adenoviral	BMP-2	Rat	Bone regeneration was similar in adipose tissue-derived MSCs and human bone marrow	[182]
		Adenoviral	BMP-2	Mice	Full bone defect regeneration was attained	[183]
		Adenoviral	BMP-2	Porcine	After 6 months complete, defect healing was obtained	[184]
		Lentiviral	BMP-2	Rat	Bone regeneration with a greater bone volume	[185]
		Retroviral	Osterix	Mouse	85% of regeneration was induced by MSCs	[186]
	In vivo	Adenoviral	BMP-2 or BMP-6	Pony	Long-term healing was not achieved for bone	[187]
		AAV	BMP-2	Mouse	A new cortical shell was formed by scAAV2.5-BMP2 allografts that was highly similar to that generated via live autografts	[188]
		Retroviral	BMP-2/4	Rat	Healing rate was similar to untreated controls and was followed by excessive formation of ectopic bone that finally remodelled	[189]
		Retroviral	COX-2	Rat	Ectopic bone formation avoided with a faster healing rate	[190]
Non-viral	Ex vivo	Gelatin	BMP-2	Rat	Homogenous bone regeneration	[191]
		Liposome	BMP-2	Porcine	Critical-size bone defects regenerated	[192]
		Liposome	Osterix	Rabbit	Thickness of new trabeculae, the amount of the newly formed bone, and bone mineral density improved	[193]
		Nucleofection	BMP2/Runx2	Mice	Bone formation was markedly higher in nucleofected cells than BMP-2 alone	[194]
		Nucleofection	BMP-6	Rat	Bone healing was greater in treated cells	[195]
	In vivo	Naked DNA	BMP-2	Mice	Increasing the number of injections led to frequent regeneration of bones	[196]
		GAM	BMP-4	Rat	After 18 weeks, an excessive amount of osteogenesis was stimulated	[197]
		GAM	PDGF	Rat	Delayed bone regeneration and prolonged inflammation	[198]
		GAM	PDGF	Rat	Improved bone healing after 4 weeks of implantation	[199]
		Sonoporation	BMP-2/BMP-7	Mouse	Ectopic osteogenic differentiation was achieved by co-delivery of genes	[200]
		Sonoporation	BMP-6	Porcine	40% of cells transfected at the fracture site. 6 weeks after treatment, complete fracture healing was observed	[201]
		Electroporation	BMP-9	Mouse	5 weeks after gene delivery led to complete repair of the bone defect	[202]
		Liposome	BMP-2	Porcine	Enhanced bone repair was monitored for treated groups	[203]

## Data Availability

Not applicable.

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
