# Peer review of "Gene Therapy for Regenerative Medicine"

_pharmaceutics, 2023, doi:10.3390/pharmaceutics15030856_

Round 1

Reviewer 1 Report

This review is very well-written. It discussed the pros and cons of viral and non-viral delivery pathways. After that, many representative cases were discussed and shown to display the feasibility of gene therapy in regenerative medicine. 

Some minor things need the author's attention. First is that table in P26 is trunked. Authors can consider making a table to summarize the pros/cons of all viral vectors to help readers quickly visualize them.

Author Response

Author's Reply to the Review Report (Reviewer 1)

We would like to thank the reviewers for their thoughtful comments and efforts towards improving our manuscript. In the following, we highlight general concerns of reviewers that were common and our effort to address these concerns. We then address comments specific to each reviewer below.

Comments: Some minor things need the author's attention. First is that table in P26 is trunked. Authors can consider making a table to summarize the pros/cons of all viral vectors to help readers quickly visualize them.

Answer: Table 1 (in the revised version, it is Table 2):

The margin of table 1 is based on the format A3 and we tried to reduce the size with format A4.

Since bone regeneration is the largest tissue that gene therapy has been used to target the specific genes (by the use viral and non-viral vectors) to regenerate the bone, we believe this table can easily compare both viral and non-viral vectors gene therapy in bone tissue engineering.

A new table 1 was added to the manuscript as Table 1 (page 20), that summarizes advantages and disadvantages of viral-medicated and non-viral mediated gene therapy.

Reviewer 2 Report

This review manuscript is very detailed and complete. Applied applications are introduced in detail, from gene therapy tools to regenerative medicine. The manuscript could be published as soon as possible after revision.

Here are some suggestions:

1. Can a paragraph describe gene therapy's risk and some bottlenecks and its applications in regenerative medicine?

2. Does the author have some more of their own comments or opinions on the relevant descriptions? Please increase them.

Author Response

Author's Reply to the Review Report (Reviewer 2)

We would like to thank the reviewers for their thoughtful comments and efforts towards improving our manuscript. In the following, we highlight general concerns of reviewers that were common and our effort to address these concerns. We then address comments specific to each reviewer below.

Comments: 1. Can a paragraph describe gene therapy's risk and some bottlenecks and its applications in regenerative medicine?

Answer: A new table 1 was added to the manuscript as Table 1 (page 20), that summarizes advantages and disadvantages of viral-medicated and non-viral mediated gene therapy.

Comments: 2. Does the author have some more of their own comments or opinions on the relevant descriptions? Please increase them.

Answer: based on the reviewer’s comment, we have added two parts of our own comments on page 19 (related to non-viral gene delivery vectors).

Also, page 32, the sentence was modified to indicate our work on bone regeneration by use of our non-viral gene vector.

Reviewer 3 Report

HHosseinkhaniA.J. Domb, and their colleagues presented a review manuscript entitled “Gene Therapy for Regenerative Medicine”. The authors focused on the recent developments and advances in regenerative medicine technologies using viral and nonviral gene delivery carriers. 

Overall, the review article is well-conceived, well-written, and worth publishing in the MDPI Pharmaceutics. However, I recommend a detailed revision addressing the following major and minor issues carefully to reach more audiences and readers of different disciplines before considering a possible publication. 

1.      Page 5, Line 118: lipoprotein lipase  lipoprotein lipase deficiency 

2.      Page 5, Line 135: The authors described the “high immunogenicity of an Ad protein”. It would be better to change Ad protein to Ad capsid. The capsid covers all proteins of adenovirus. Note that the adenoviral capsid surrounds the viral core and is composed of major (penton, hexon, fiber) and minor (pIX, pIIIa, pVI,and pVIII) capsid proteins

3. Redundant abbreviations or acronyms were used throughout the manuscript. 

Line 131, Line 224, and Line 274: inverted terminal repeats (ITRs) 

 Line 449, and Line 454: small interfering RNA (siRNA) 

4.      Provide the full description of the abbreviation or acronym after its first appearance in the manuscript. 

           Line 138: HIV  human immunodeficiency virus (HIV)

Line 558: FGF-1  fibroblast growth factor-1 (FGF-1) [Later, in line 566, adenoviral fibroblast growth factor (FGF)-4, the authors can use adenoviral FGF-4]

Page 28: NFκB  Nuclear factor kappa B 

Figure 10: PDL  periodontal ligament

Figure 10: hGFs  human gingival fibroblasts (hGFs) 

5.      Unnecessary abbreviations or acronyms were used, even though the description was used only once. They will distract the reader's attention. 

      For example: 

      Line 316: self-inactivating (SIN)  self-inactivating

      Line 519: blood-brain barrier (BBB)  blood-brain barrier

     Page 28: (RANKL) 

Page 38: (cAMP) 

Page 44: (HeCs), (DCs), and (OPCs)

Page 51: (GelMA) 

Page 56: (FA-PEG

Page 56: (MSNs) 

6.      Line 430: LipofectinTM  Lipofectin™️

7.      Line 431: Conventional phosphate is another method for the formation of DNA nanoparticles. This line is misleading and incomplete. Do the authors mean calcium phosphate co-precipitation?

8.      Line 438: No references were provided to the cell receptor-mediated uptake. 

9.      Line 460: The authors mentioned that “perfect mRNA delivery-based gene therapy is required to prevent serum endonucleases”. Here, the author’s description is limited to only serum and endonucleases. However, ribonucleases are omnipresent and are of two different types endoribonucleases and exoribonucleasesHence, I recommend that the authors change the description with proper citations as follows. Perfect mRNA delivery-based gene therapy mRNA is required to prevent degradation by extracellular and intracellular endo- and exo-ribonucleases (J Drug Target 2019;27(5-6):670-680 and Mol Pharm 2018;15(6):2268-2276)

10.  Expand the full form of cDNA. 

11.  Table 1: Not the whole table is shown. It may be a technical issue. Please provide the full table. 

12. Page 54: 3.2.2. Scaffolds-mediated regeneration. No discussion/information was provided under section 3.2.2.   

13.  Line 513: coronary heart disease (CAD  coronary heart disease (CHD)

14.  After cellular entry by endocytosis, the endosomal escape of mRNA is one of the major bottlenecks of successful mRNA-based gene therapy. Hence, I recommend that the authors introduce endo/lysosomal escape in the following sentence. 

Perfect mRNA delivery-based gene therapy mRNA is required to prevent degradation by extracellular and intracellular endo- and exo-ribonucleases, escape immune detection (Immunity 2005;23(2):165-75 PLoS One 2013;8(2):e56220), endo/lysosomal escape (Macromol Rapid Commun 2022;43(12):e2100754J Cell Biol 2022;221(2):e202110137), avoid non-specific interactions with non-target cells or biomolecules, prevent renal clearance, permit extravasation to reach tissues of interest, and enhance cell entry (Nat Rev Mater 2, 17056 (2017)). 

15.         3The Center for Nanoscience and Nanotechnology  2The Center for Nanoscience and Nanotechnology 

Author Response

please find in the word file response to the reviewer’s comments

Reviewer 4 Report

The paper of Hosseinkhan et al. is a huge paper dealing with gene therapy in regenerative medicine. The paper  is interesting but I have some questions to the authors.

The paragraph 2 (Gene therapy technology) is too long, is reporting topics that are already known, sounds heavy and redundant and unnecessary, and could be deleted. The real review is starts from the paragraph 3.

Minor points

Line 28: missioning is not the right word

Line 237: risk instead of isk

Line 300 Pompe name shoud have the capital letter

Line 431 Conventional phosphate? Maybe  Calcium phosphate

Line 483 not clear to me “identification of desidered tissues”

Line 500 cointaining?

Table 1 is not correctly shown an is not necessary

Paragraph 3.7.1 The first two sentences are not clear to me. How gene therapy can potentially infect cochlear hair cells?

Please check the english

Author Response

please find in the uplaod word file  response to the reviewer’s comments 

Round 2

Reviewer 3 Report

Accepted in present form

Author Response

We would like to thank the reviewers for their thoughtful comments and efforts towards improving our manuscript and accepted our revised manuscript.

we also further checked grammar and English editing of our paper.

Reviewer 4 Report

The paper improved.

Author Response

(The authors gave the same response as above.)
